# Free-Resolution Probability Distributions Map-Based Precise Vehicle Localization in Urban Areas

**DOI:** 10.3390/s20041220

**Published:** 2020-02-23

**Authors:** Kyu-Won Kim, Gyu-In Jee

**Affiliations:** Department of Electrical and Electronic Engineering, Konkuk University, 120 Neungdong-ro, Gwangjin-gu, Seoul 05029, Korea; kkw1125@konkuk.ac.kr

**Keywords:** free-resolution probability distributions map (FRPDM), precise vehicle localization, 3D LIDAR, urban area, road marking, vertical structure

## Abstract

We propose a free-resolution probability distributions map (FRPDM) and an FRPDM-based precise vehicle localization method using 3D light detection and ranging (LIDAR). An FRPDM is generated by Gaussian mixture modeling, based on road markings and vertical structure point cloud. Unlike single resolution or multi-resolution probability distribution maps, in the case of the FRPDM, the resolution is not fixed and the object can be represented by various sizes of probability distributions. Thus, the shape of the object can be represented efficiently. Therefore, the map size is very small (61 KB/km) because the object is effectively represented by a small number of probability distributions. Based on the generated FRPDM, point-to-probability distribution scan matching and feature-point matching were performed to obtain the measurements, and the position and heading of the vehicle were derived using an extended Kalman filter-based navigation filter. The experimental area is the Gangnam area of Seoul, South Korea, which has many buildings around the road. The root mean square (RMS) position errors for the lateral and longitudinal directions were 0.057 m and 0.178 m, respectively, and the RMS heading error was 0.281°.

## 1. Introduction

Precise vehicle localization must be performed for autonomous driving, wherein a localization accuracy of less than 0.5 m in the lateral direction [1] and less than 1 m in the longitudinal direction is generally required for a 95% confidence level. A global navigation satellite system (GNSS), such as the global positioning system (GPS) used for vehicle localization, utilizes the satellite’s outgoing radio waves to determine the position. However, in an environment with many buildings, such as an urban area, only a small number of satellite radio waves are received because many waves are blocked. Because of the multipath error caused by reflection of radio waves by the buildings, the above conditions cannot be met, even with precision navigation equipment such as a GPS real-time kinematic (RTK).

To solve this problem, many studies are being conducted to generate a map using various sensors and to match it with the sensor data to perform vehicle localization. Especially, 3D light detection and ranging (LIDAR) is widely used because it provides precise distance and intensity information using a 3D point cloud. 3D-LIDAR-based maps largely include point cloud maps, line maps, landmark maps, grid maps, and probability distribution maps.

A point cloud map [2,3] is a map that accumulates the scanned LIDAR point cloud and uses the LIDAR scan data as they are, enabling precise position estimation. However, because it stores a large amount of data, it requires a very large map size and takes a long time to process.

A line map [4,5,6] detects road markings or vertical structures in LIDAR point clouds and converts them into lines. Because only lines are stored in the map, the map size is small. However, the position estimation performance is relatively low. In addition, such maps consist of lines, errors may occur depending on the positions of the detected lines because the position of the line to be detected may vary depending on the width. Therefore, large errors may occur when matching with a line of a wide-width object.

A landmark map [7,8] is generated by detecting a specific landmark. Because only the landmark points are stored in the map, the map size is very small. In addition, if the landmark detection performance is good, the localization performance is also outstanding. However, when a landmark does not exist, or is insufficient, the vehicle localization performance is very poor. Because a landmark must be detected, position errors may occur because of misdetection.

A grid map [9,10,11] is a map of the surrounding environment divided into grids of a certain resolution. Because the scan data is created as a grid map for matching, no detection process is required. Fast matching is possible with its form of a two-dimensional (2D) matrix by using the fast Fourier transform correlation method, as shown in [11]. However, a larger grid resolution degrades the localization performance, and a smaller resolution increases the map size.

Lastly, a probability distribution map (PDM) [12,13] is a map of LIDAR point cloud data expressed as the probability distribution. In general, it is generated as a map based on a single Gaussian probability distribution. The point cloud is divided into grids of a single resolution to derive the mean and covariance of each grid. Because the shape of the point cloud is expressed using the probability distribution, the map size is small. If the probability distribution is similar to the shape of the point cloud, the map matching performance is outstanding. Unlike iterative closest point (ICP) scan matching [14], which is a point-to-point matching method, the processing speed is faster because a relatively smaller number of probability distributions and matchings are performed. Normal distributions transform (NDT) scan matching is a commonly used matching method [15,16]. NDT scan matching derives the probability density function (PDF) between the point cloud and the probability distribution in the map grid, and the optimal solution is derived through the generated score function using a nonlinear function optimization method. Because it is performed iteratively until it finds the minimum value of the function, the processing speed is determined by the speed of convergence. However, for a single resolution based PDM, the matching performance and processing speed vary, depending on the grid resolution. If grid resolution is large, matching is faster because of the faster convergence speed. In contrast, if there is a complex point cloud, it may not be expressed properly, resulting in poor matching performance. If the grid resolution gets smaller, complex shapes can be represented, but the map size increases because of the large number of grids. Because of the low convergence speed, the matching speed is also reduced. In addition, because there is a high possibility of falling into local minima, it is difficult to correct the error again when the position error increases.

In this study, a free-resolution probability distributions map (FRPDM) was generated using the Gaussian mixture modeling (GMM) and proposed as an accurate and robust vehicle localization method for urban areas. The FRPDM does not have a fixed resolution, unlike a multi-resolution map that consists of multiple fixed grid resolutions. Therefore, it can be expressed as the probability distribution of various sizes according to the size of the object.

The existing multi-resolution based PDMs include the Gaussian mixture map proposed by Wolcott [17,18] and the multi-layered normal distributions transform (ML-NDT) [19] proposed by Ulaş. Wolcott generated the multi-resolution Gaussian mixture map by the method in which the height of the building scanned with LIDAR is converted into the probability distribution for each grid and stored in the smallest grid, and the probability distributions of smaller grids are incorporated into that of a larger resolution grid using GMM. Using this method, the root mean square error (RMSE) is estimated to be very precise, at 0.1 m and 0.13 m, in the lateral and longitudinal directions, respectively. However, because the search area is set and the correlation is performed by grid resolution, the processing speed is much less (4500 ms, CPU only), and the map size is relatively large (44.3 MB/km). Ulaş proposed a scan matching technique using multi-resolution ML-NDT by setting multiple grid resolutions. For the ML-NDT, the problem of the convergence speed and matching performance determined according to the grid size in the existing NDT scan matching was resolved by generating the probability distributions of grids of various resolutions and performing scan matching sequentially from the largest grid resolution. However, the processing speed is less because scan matching is required for each grid resolution. Because the probability distribution is stored for each grid resolution, the map size is large as well.

In contrast, the FRPDM divides the extracted point cloud by object without any resolution and converts it into probability distributions through GMM. Because multiple probability distributions can be generated for each object, the shape of the object can be represented effectively in the probability distribution of various sizes.

Figure 1 shows a comparison of the PDM generation results, where panel (a) is the 3D LIDAR point cloud and panels (b) and (c) show the results of generating a single resolution PDM with grid resolutions of 2 m and 1 m, respectively. As shown in this figure, panel (b) does not properly express the form of road markings. In contrast, panel (c) is almost the same as the road markings. However, because there are a large number of probability distributions, the map size increases. In contrast, panel (d), which is generated by the FRPDM, effectively expresses the form of road markings with a small number of probability distributions. In addition, unlike the existing single resolution PDM, the FRPDM involves probability distributions of various resolutions, enabling fast and precise map matching with one scan matching.

In addition, road markings and vertical structures were extracted from the LIDAR point cloud to generate the FRPDM. Road markings are always present on the road. Therefore, they are mainly used as the information for vehicle localization. In addition, in urban areas, there are many vertical structures around the roads. A vertical structure refers to a structure built vertically on the ground, such as a building or a traffic sign, around the road. This factor causes a GNSS error but can also be used as information for localization. When localization is performed using road markings, it can be used in most cases because road markings always exist on the road. Because they are placed on the road precisely, precise vehicle localization can be performed. However, if there are many obstacles on the road, localization performance is reduced because there are few road-marking points that can be extracted. In the case of a large position error, it is difficult to perform data association. In contrast, vertical structures are less distributed around roads than road markings. Because the outer wall is long, it is easy to perform data association. Even if a large position error occurs, it is easy to correct the error. However, because of the small distribution, it is difficult to derive a precise localization performance. Therefore, if road markings and vertical structures are used at the same time, their advantages are combined to offset their disadvantages, and to obtain a very precise vehicle localization result.

Figure 2 shows the process of vehicle localization. As shown in Figure 2, we conducted error correction of the initial vehicle position and heading. When error correction was completed, we performed a time update step using the position and heading increment from the GPS/dead reckoning (DR). Finally, we obtained measurements from map matching, and the position and heading were estimated using the navigation filter with measurements.

The contributions of this study are as follows:Proposal of a new type of map generation technique called the FRPDM,Proposal of the precise vehicle localization method in urban areas based on the FRPDM map.

The proposed FRPDM has the following advantages:We generate an easily available map in urban areas using road markings and vertical structures,We generate a map with a very small size (61 KB/km),Automatic map generation is performed through algorithm execution without manual work,We satisfy the localization accuracy requirements for autonomous vehicle driving,We perform fast map matching (average 37 ms, CPU only),Even if the initial position error is large, error correction is possible.

This paper is organized as follows. Section 2 describes how to generate the FRPDM, and Section 3 explains the FRPDM-based precise vehicle localization method. Section 4 analyzes the FRPDM-based vehicle localization performance through experimental results, and Section 5 concludes this study.

## 2. Method of Free-Resolution Probability Distributions Map (FRPDM) Generation

In this section, we explain how to generate the FRPDM. The FRPDM is a map that extracts road markings and vertical structures from the 3D LIDAR point cloud and stores the probability distributions converted through the GMM. Unlike conventional single resolution or multi-resolution maps, the FRPDM has no fixed resolution, and the probability distribution depends on the size of the object. In addition, one object can be represented by multiple probability distributions using the GMM. Therefore, even a complex object can be represented by multiple probability distributions. This allows us to generate a map that efficiently represents the object form with a small map size.

Figure 3 shows a flowchart of the FRPDM generation process. As shown in this figure, the FRPDM is generated through three processes: generating the pose graph optimization based on precise map trajectory [20,21,22], extracting the road-marking/vertical structure point clouds [6], and generating the GMM based on the probability distribution map. The detailed FRPDM generation method is described below.

### 2.1. Precise Map Trajectory Generation Based on the Pose Graph Optimization

A precise map trajectory derivation is essential for vehicle localization using a map. Generally, an expensive GPS (RTK)/inertial navigation system (INS) is used to derive a trajectory for the mobile mapping system (MMS) [23,24]. However, when there are many buildings around, such as in urban areas, errors occur because of multipath even when expensive equipment is used.

Figure 4 shows the map trajectory of an urban area for the FRPDM generation. As shown in this figure, buildings are concentrated in all sections. Therefore, inconsistency occurs in the map trajectory because of multipath error, thus negatively impacting the reliability of the generated map. To solve this problem, we performed pose graph optimization. GPS (RTK)/INS equipment (NovAtel RTK/SPAN system) was used for the initial trajectory for pose graph optimization. The edge measurements for graph generation were used to obtain the relative position and posture between the trajectory nodes, derived by the ICP scan matching based on the point cloud scanned with 3D LIDAR.

Figure 5 shows the result of pose graph optimization. On the left are road-marking maps [11], generated based on trajectories derived by the GPS (RTK)/INS and the pose graph optimization, respectively. On the right is the result of each trajectory. While the road markings generated by the GPS (RTK)/INS trajectory are inconsistent with each other, the road markings generated by the pose graph optimization match each other. In this study, we generated the FRPDM based on the trajectory derived from the pose graph optimization and used it as the ground truth for analyzing the position estimation performance.

### 2.2. Extraction of Road Markings and Vertical Structures

To generate a probability distribution map using the LIDAR point clouds, point cloud extraction is required to generate probability distributions. In this study, we extracted the point clouds of the road markings and vertical structures to generate a probability distribution map. Road markings were painted on the road using bright paint to make them stand out from the surrounding asphalt. Consequently, the road markings could be distinguished from the asphalt using the intensity differences. The vertical structures mainly comprise buildings and traffic signs in the form of lines, so they can be separated by the line extraction algorithm. The detailed description is as follows.

#### 2.2.1. Road-Marking Extraction

For road-marking extraction, it is necessary to first select the point clouds on only the road. For this study, only the point clouds on the road were extracted using the height filter [25]. To use the height filter, only the point clouds below a certain height were selected for each LIDAR layer. The height differences between the consecutive points among the selected point clouds were compared to distinguish the point clouds with the greatest height difference and extract the point clouds that were the lowest. Through this process, the dynamic objects on the road surface were removed, and only the road point clouds were extracted.

Figure 6 shows the result of road point cloud extraction through the height filter. The red point clouds are the extraction results, and the blue point clouds are the other point clouds. As shown in this figure, only the road surface is extracted and the surrounding dynamic objects are removed.

To extract the road markings from the extracted road point clouds, the binarization process using LIDAR intensity was required. For this purpose, we used the Otsu thresholding method [26]. This method finds the optimal threshold by using the brightness distribution of the image pixels. It is possible to find the optimal threshold with this method even if the intensity level changes.

Figure 7 shows the result of road-marking extraction. As shown in this figure, only the road-marking points were extracted apart from the asphalt. All road-marking point clouds extracted for the FRPDM generation were accumulated and stored.

#### 2.2.2. Vertical Structure Extraction

To extract the vertical structure, we performed line fitting for each LIDAR layer using the iterative-end-point-fit (IEPF) algorithm [27]. The IEPF algorithm performs clustering by comparing the distance between the points and fitting the clustered points to a line through the split–merge process. Compared with several line fitting algorithms, this method exhibits high processing speed and suitable accuracy [28].

Figure 8 shows the result of line fitting using the IEPF algorithm. IEPF was performed by extracting the point clouds above a certain height for line extraction. All the point clouds were fitted into the lines. However, as shown in this figure, there are not only the vertical structures, but also outliers such as roadside trees. Therefore, an outlier removal process was required.

Figure 9 shows the pseudocode of the outlier removal algorithm. Outlier removal was performed using the variance of the line and the length of the line. The variance of the line denotes the distances between the lines and the points. First, for outlier removal, the lines fitted for each layer were combined through the clustering process. Then, the outliers were removed by selecting only the lines satisfying the conditions, using the variance of the line and the line length.

Figure 10 shows the result of the vertical structure line extraction. Blue circles denote outliers, and the vertical structures extracted are indicated as red dots. As shown in this figure, the outliers were removed, and only the vertical structures were extracted.

The extracted road markings and vertical structures were accumulated according to the map trajectory nodes, as shown in Figure 11. The probability distribution map was generated using all the accumulated point clouds.

### 2.3. Gaussian Mixture Modeling (GMM)-Based FRPDM Generation

To generate the FRPDM using the road-marking and vertical structure point clouds extracted in Section 2.2, the point cloud must be converted into probability distributions. The conversion of the point clouds into probability distributions comprises four steps: eliminating the outlier points using the occupancy grid filter, clustering the point clouds through object clustering, generating probability distributions using the GMM, and removing overlapped probability distributions.

Although outliers were removed when the road-marking and vertical structure point clouds were extracted, as stated in Section 2.2, a small number of outliers may still need to be extracted. To eliminate these, we used an occupancy grid filter [29]. The occupancy grid filter removes the point clouds with low probability from the grid. This is done by dividing the point clouds into grids of a certain size and calculating the occupancy probability per grid. Generally, because only a few outlier points tend to be extracted, the occupancy probability is calculated to be low. Therefore, they can be removed by the occupancy grid filter.

Figure 12 shows the result of the outlier removal using the occupancy grid filter. As shown in this figure, the outliers are removed by the occupancy grid filter because of the low occupancy probability when the occupancy grid is generated. After removing the outliers through the occupancy grid filter, the point clouds are classified via object clustering.

Gaussian mixture probability distribution conversion using the GMM was performed in an entire area of the map for representing the whole shape of the object. Because the probability distribution is estimated using the maximum likelihood estimation (MLE) method, the processing time exponentially increases when all point clouds are converted at once. Therefore, the processing time can be reduced by classifying the point clouds by object through clustering and performing the GMM for each object.

Figure 13 shows the result of object clustering. Clustering was performed using the k-nearest neighbor algorithm. As shown in Figure 13, the point clouds were classified by object and were converted into the probability distributions through the GMM. A method called the expectation–maximization (EM) algorithm [30] can be applied for the GMM. 

Figure 14 shows the pseudocode of the EM algorithm for the GMM. Point clouds classified by object clustering and the number of probability distributions are input. First, the K-mean clustering is conducted for the parameter initialization. After the parameter initialization, the expectation-step (E-step) and maximization-step (M-step) are performed repeatedly to derive the parameters of the probability distribution. As shown in this figure, the likelihood estimation is performed by the E-step, and the parameters of the probability distribution are determined by the M-step.

However, to perform GMM, it is necessary to determine the number of probability distributions to be estimated. Therefore, we designed a method of increasing the number of probability distributions until a certain threshold was reached, after calculating the ratio as the density of the probability distribution and the point cloud therein.

Figure 15 shows an example of increasing the number of probability distributions. As shown in the figure on the left, when the number of probability distributions is one, the probability distribution is generated by the distribution of the entire point cloud. At this time, the ratio can be derived by calculating the density between the probability distribution and the point cloud. The equation is as follows.
(1)$$\begin{array}{c} \left[\begin{array}{cc} e_{1} & e_{2} \end{array}\right]=2\cdot \left[\begin{array}{cc} v_{1} & v_{2} \end{array}\right]\\ \rho_{PD}=n^{2}/(e_{1}\cdot e_{2})\\ \rho_{PC}=N/(e_{1}\cdot e_{2})\\ r=\rho_{PC}/\rho_{PD} \end{array}.$$

Equation (1) calculates the density between the probability distribution and the point cloud. First, to calculate the density, it is necessary to uniformize the distribution of point clouds. Therefore, we performed down sampling to distribute the point clouds at regular intervals using the grid filter prior to performing the GMM. v is the Eigen vector of the estimated probability distribution, and e is the length of probability distribution. We can derive the area of the probability distribution using $e_{1}$ and $e_{2}$. The density of the probability distribution ($\rho_{PD}$) is derived by dividing the square of (n), which is the inverse of the grid size, by the area of the probability distribution. Likewise, the density of the point cloud ($\rho_{PC}$) can be derived by dividing the number of point clouds in the probability distribution (N) by the area of the probability distribution. The ratio between the two densities (r) is calculated, and if it is less than the threshold, the number of probability distributions is increased until the condition is satisfied, as shown in Figure 15. If all point clouds in the object satisfy the condition, the probability distribution estimation ends.

When the GMM is completed, the point cloud is converted into the probability distribution (Figure 16). As shown in the figure, the road-marking point clouds were converted into the probability distributions with similar shapes. Likewise, complex shapes can be efficiently represented in probability distributions through the GMM.

When the generation of the probability distributions using the GMM is completed, overlapped probability distributions should be removed. Because the probability distributions generated above are determined according to the number of probability distributions, there is a possibility that another probability distribution overlaps within the probability distribution. If there is another probability distribution that overlaps with the generated probability distribution, unnecessary calculations may have to be performed when matching the map. In addition, the map size increases as the number of probability distributions increases.

Figure 17 shows the pseudocode for the removal of the overlapped probability distributions. First, the Mahalanobis distance between two probability distributions (ζ) is derived, which is the distance between two points in terms of the ratio of standard deviation. Therefore, it is the most optimal parameter that can determine if a probability distribution exists in another probability distribution. If ζ is less than the threshold, it is determined that the mean of a probability distribution exists in another arbitrary probability distribution. At this time, the eigenvalues of the two probability distributions are derived. The eigenvalue refers to the size of covariance. Therefore, by comparing the eigenvalues of the two probability distributions, we can determine which probability distribution is larger. If the ratio of the eigenvalues between two probability distributions is greater than a certain threshold, the probability distribution of the smaller eigenvalue is removed. Through this process, smaller probability distributions existing in a probability distribution are eliminated.

However, in addition to the above cases, there are overlapping probability distributions. To remove them, the Mahalanobis distance for the remaining probability distribution is calculated. If ζ is less than the threshold, the correlation coefficient of the two probability distributions (ρ) is derived. The correlation coefficient is a parameter representing the degree of correlation between the covariance variables, which can be used to find the direction of probability distribution. If the difference between the correlation coefficient of the two probability distributions is less than a certain threshold, it can be determined that the two probability distributions almost overlap. Through this process, the overlapped probability distributions are removed, resulting in the final map.

Figure 18 shows the result of overlapped probability distribution removal. In Figure 18a, the smaller probability distribution inside a probability distribution is removed. In Figure 18b, the probability distributions overlap with each other. Even when the probability distributions are overlapping, they are removed using the correlation coefficients.

Table 1 is an example of the FRPDM. As shown in this table, the type of probability distribution, the mean of the probability distribution ($\mu_{x}$ and $\mu_{y}$), and the covariance elements of the probability distribution ($\sigma_{x^{2}}$, $\sigma_{xy}$ and $\sigma_{y^{2}}$) are stored in the map. As shown above, because there is little information stored in the map, the capacity required to generate the map is very small. The mean of the probability distribution is stored as the absolute position coordinates, based on the latitude and longitude, so that it can be used for any vehicle and the covariance of the probability distribution is stored as the ENU frame.

### 2.4. Advantages of Using the FRPDM

The FRPDM has two advantages over other maps. The first is that the map size is extremely small, and the second is the ability to effectively represent objects of various sizes.

As shown in Table 1 in Section 2.3, the FRPDM stores only six elements in the map. The FRPDM has no fixed grid resolution, and the size of the probability distribution depends on the size of the object. Therefore, even large objects can be represented by one probability distribution without division, so the map size is very small compared to other maps.

Table 2 shows a comparison of the map sizes per km. As shown in this table, the FRPDM is the smallest in size. The smaller the map size, the wider the area a map can cover. Therefore, it can be regarded as a suitable map when considering a vehicle moving over a large area.

The FRPDM provides sufficient information for vehicle localization even when the map size is small. In the example shown in Table 2, the multi-resolution Gaussian mixture map stores the height of the vertical structure as Gaussian mixture distributions. Therefore, if there are few buildings around, the localization performance is low, and the availability is low as well. It also has the largest size among the maps in Table 2. In the case of a binary grid map, only binary information is included in the grid through the binarization process. Therefore, the map size is smaller than a general grid map. However, because the grid resolution must be small for precise localization, the basic map size is relatively large as compared to other maps. The extended line map is a map that stores road markings and vertical structures as lines. Because it only stores the node links of the lines, the map size is small. However, because of the nature of a line, only the length is expressed, which causes errors depending on the width of the object. The probability distribution map can express not only the length but also the width of objects using the mean and the covariance. Therefore, the localization performance may be the best when the shape of the object is similar to that of the probability distribution. However, the single resolution-based PDM expresses the shape of an object with a single Gaussian probability distribution, depending on the grid resolution. Therefore, if the shape of the object is complicated, it is impossible to express it properly with a large resolution. As the resolution decreases, the map size increases exponentially. In contrast, the FRPDM expresses an object with multiple probability distributions using the GMM. Because it does not have a fixed resolution, an object can have probability distributions of various sizes. Therefore, even complex shapes, such as road markings, can be effectively expressed, thereby improving the localization performance.

## 3. FRPDM-Based Precise Vehicle Localization Method

In this section, we explain the precise vehicle localization method using the FRPDM generated in Section 2. For the FRPDM, the size of the probability distribution depends on the size of the object without a fixed resolution, as described in Section 2. Consequently, because there are probability distributions of various sizes, it is possible to provide fast and precise map matching, unlike the traditional single resolution PDM, where the map matching speed and performance depend on the grid resolution. For the multi-resolution PDM, precise map matching is possible by matching probability distributions of multiple resolutions; however, the processing is slow because it must process all of the grid resolution. In contrast, the FRPDM performs map matching only once, so it can be processed quickly.

Figure 19 shows the precise vehicle localization process based on the FRPDM. The sensors used for vehicle localization are the GPS/DR and the 3D LIDAR. Vehicle localization is performed through map matching between the FRPDM and the accumulated point clouds that are extracted by the road-marking and vertical structure extraction algorithms described in Section 2.2. Map matching is performed by the point-to-probability distribution scan matching and feature–point matching methods. The vehicle position and heading are estimated through the extended Kalman filter (EKF)-based navigation filter using the measurements acquired by map matching. When we start vehicle localization, the GPS/DR position is used for initial vehicle position. However, if the position error of the GPS/DR is large, map matching is very hard because sensor data cannot be associated with the map. To solve this problem, we perform error correction of the initial vehicle position for map matching when we start the vehicle localization process.

### 3.1. Map Matching Algorithm

In the case of the FRPDM, the grid resolution is not fixed, unlike the single resolution PDM or multi-resolution PDM. Therefore, a new algorithm is required for point-to-probability distribution scan matching. In this study, we propose a data association method between point and probability distribution for scan matching based on the FRPDM. In addition, we propose feature point map matching based on probability distribution. Using data association of the point-to-probability distribution process, we can transform the point cloud to a probability distribution. After that, we perform data association between the probability distribution of the map and that of the point cloud. As a result, we can extract feature points that match the probability distribution of the map.

The measurements for vehicle localization are obtained by matching the FRPDM with the point clouds of the road markings and vertical structures extracted by the 3D LIDAR. However, there may not be sufficient information for map matching using only the point clouds at the current time. Because the road markings are painted on the road, only some of the road markings may be extracted when there are many obstacles around, such as the vehicles during a traffic jam. In addition, because the 3D LIDAR has a wide interval between vertical layers, information for longitudinal estimation is relatively insufficient. Therefore, the point cloud of the previous time point is accumulated using the sliding window method to compensate for the insufficient longitudinal information [31].

Figure 20 outline the process of point cloud accumulation. The GPS/DR is used for point cloud accumulation of the position and heading per time point. As shown in this figure, the point clouds are accumulated to compensate for the insufficient longitudinal information.

Figure 21 shows the pseudocode of the point-to-probability distribution scan matching. The point-to-probability distribution scan matching obtains the score and the Jacobian function between the point and the probability distribution and then derives the optimal state variables through nonlinear optimization (like the conventional NDT scan matching [15,16,19]), where the state variables, (x), are the results of error correction through scan matching between the map and the point cloud. However, unlike the NDT scan matching, which can be directly matched with probability distributions in a grid, the FRPDM does not have a fixed grid, and a data association process is required for matching between the point and the probability distribution.

Data association is performed through the process shown in Figure 22. For data association, the probability distributions of the same type as the point are found, where the type means whether the probability distribution of the map is a road marking or a vertical structure. After finding the probability distribution of the matching type in the map, the Euclidian distance between the point and the probability distribution is derived. The Euclidian distance is the actual distance between two points, which gives the distance between the point and the mean of the probability distribution. By comparing the Euclidean distances, we can find the nearest probability distribution. However, even if the actual distance is less, the probability distribution may not match with the actual point because the magnitude of each probability distribution is different. Therefore, the Mahalanobis distance ($d_{MH}$) between the point and the probability distribution found by the Euclidean distance is derived. As explained in Section 2.3, the Mahalanobis distance confirms whether a point is included in the probability distribution. However, even though the actual point and the probability distribution coincide because of a localization error, the point may exist outside the probability distribution. Therefore, $\Sigma_{ROI}$ is added as the region of interest (ROI) to the covariance of the probability distribution to derive $d_{MH}$. As a result, if $d_{MH}$ is less than the threshold, the probability distribution that matches is finally found.

Once we obtain the probability distribution corresponding to the point through data association, we can derive the score and the Jacobian between the point and the probability distribution.
(2)$$\begin{array}{l} m=M_{idx_{match}}\{\mu ,\Sigma \}\\ p={\hat{P}}_{i}\\ q=(p-\mu )\\ \frac{\partial q}{\partial x}=\left[\begin{array}{ccc} \frac{\partial q_{x}}{\partial x_{t_{x}}} & \frac{\partial q_{x}}{\partial x_{t_{y}}} & \frac{\partial q_{x}}{\partial x_{\theta }}\\ \frac{\partial q_{y}}{\partial x_{t_{x}}} & \frac{\partial q_{y}}{\partial x_{t_{y}}} & \frac{\partial q_{y}}{\partial x_{\theta }} \end{array}\right]=\left[\begin{array}{ccc} 1 & 0 & -p_{x}\sin\theta -p_{y}\cos\theta \\ 0 & 1 & p_{x}\cos\theta -p_{y}\sin\theta \end{array}\right]\\ S(p,m)=exp(-\frac{1}{2}q^{T}\Sigma^{-1}q)\\ J(p,m)=q^{T}\Sigma^{-1}\frac{\partial q}{\partial x}exp(-\frac{1}{2}q^{T}\Sigma^{-1}q) \end{array}.$$

Equation (2) is used to derive the score and the Jacobian between the point and the probability distribution [15]. The score between the matched point and the probability distribution (S(p,m)) is derived through data association. The Jacobian (J(p,m)) is derived by the partial differentiation of the score.
(3)$$\begin{array}{c} S(x)={\left[\begin{array}{cccc} S_{1} & S_{2} & \cdots & S_{N} \end{array}\right]}^{T}\\ J(x)={\left[\begin{array}{cccc} J_{1} & J_{2} & \cdots & J_{N} \end{array}\right]}^{T}\\ D(x)=diag(J{\left(x\right)}^{T}\cdot J(x))\\ \delta (x)=-{\left(J{\left(x\right)}^{T}J(x)+\lambda D(x)\right)}^{-1}J{\left(x\right)}^{T}S(x) \end{array}.$$

Equation (3) generates the score function (S(x)) and the Jacobian function (J(x)) using the score and the Jacobian derived for each point and derives the result (δ(x)) using the Levenberg–Marquadt optimization method [19]. The Levenberg–Marquardt optimization method converges faster than the Gauss–Newton method used for the general NDT scan matching and finds the optimal solution more reliably. This reduces the processing speed by finding solutions through fewer iterations. Therefore, the point-to-probability distribution scan matching was performed using the Levenberg–Marquadt optimization method.

Figure 23 shows the result of point-to-probability distribution scan matching. We generate the single resolution PDM (2 m) and the FRPDM based on a reference point cloud. We perform map matching between initial point cloud and probability distribution using the single resolution PDM and the FRPDM. As shown in Figure 23, when scan matching is performed using the single resolution PDM, the error between reference point cloud and scan matching result is larger than that between the reference point cloud and the initial point cloud. In contrast, the scan matching result using the FRPDM almost matches the reference point cloud. The FRPDM can similarly express complicated shapes of point cloud using multiple probability distributions. However, the single resolution PDM has only one probability distribution in the grid. Therefore, the complicated shape of the point cloud cannot be represented by the probability distribution properly. As a result, scan matching is not good because the correlation between the point cloud and probability distribution is ambiguous.

Other than the point-to-probability distribution scan matching, the measurements for vehicle localization can be derived through the feature–point matching using the FRPDM. For using the feature–point matching, we need to define the feature–point. The feature–point used in this study was the mean of the probability distribution, and data association was performed using the covariance. For the data association between the FRPDM and the point clouds, the point clouds need to be converted into probability distributions.

Figure 24 shows an example of the point cloud to probability distribution conversion. As shown on the left, point cloud matching with the probability distribution of the map can be found through the point-to-probability distribution data association. Then, the matched point cloud is converted to a probability distribution, as shown on the right, to enable data association with the probability distribution of the map.

The data association between probability distributions was performed as shown in Figure 25. There are three parameters for data association between probability distributions: the distance, the correlation coefficient, and the eigenvalue between the probability distributions. The distance between probability distributions was derived using the mean of the probability distributions. The correlation coefficient and the eigenvalue were derived using covariance. The correlation coefficient and the eigenvalue indicate the direction and magnitude of the covariance, respectively. Therefore, this can be used to determine the similarity between two probability distributions. In addition, the distances between the probability distributions were compared to minimize errors that may be caused by incorrect data association.

### 3.2. Extended Kalman Filter (EKF) Configuration

In this section, we explain how to estimate vehicle position and the heading, using the navigation filter configuration [32] based on the EKF. To use the EKF, the state and input vector need to be defined.
(4)$$\begin{array}{c} X_{t}={\left[\begin{array}{ccc} x_{t} & y_{t} & \theta_{t} \end{array}\right]}^{T}\\ u_{t}={\left[\begin{array}{ccc} \Delta x_{DR_{t}} & \Delta y_{DR_{t}} & \Delta \theta_{DR_{t}} \end{array}\right]}^{T} \end{array}.$$

Equation (4) defines the state and the input vector. The vehicle position in the ENU frame is denoted by $x_{t}$ and $y_{t}$. $\theta_{t}$ is the heading of the vehicle. The input vector is the increment of the position and the heading of the GPS/DR. The time update using the state and the input vector is as follows.
(5)$$\begin{array}{c} {\hat{X}}_{t+1}^{-}=F{\hat{X}}_{t}+Gu_{t}\\ P_{t+1}^{-}=FP_{t}F+Q\\ F=\left[\begin{array}{ccc} 1 & 0 & 0\\ 0 & 1 & 0\\ 0 & 0 & 1 \end{array}\right]\;G=\left[\begin{array}{ccc} 1 & 0 & 0\\ 0 & 1 & 0\\ 0 & 0 & 1 \end{array}\right] \end{array}.$$

In Equation (5), the state of the vehicle is updated using the increment of the GPS/DR.
(6)$$\begin{array}{c} r_{i}=\sqrt{{\left(x_{i}-x_{t}\right)}^{2}+{\left(y_{i}-y_{t}\right)}^{2}}\\ \beta_{i}=tan^{-1}(\frac{y_{i}-y_{t}}{x_{i}-x_{t}})-\theta_{t}\\ z=h(X_{t},m)=\left[\begin{array}{c} x_{t}+\Delta x_{t}\\ y_{t}+\Delta y_{t}\\ \theta_{t}+\Delta \theta_{t}\\ r_{1}\\ \beta_{1}\\ \vdots \\ r_{n}\\ \beta_{n} \end{array}\right] \end{array}.$$

Equation (6) defines the measurement for the measurement update. Measurement z can be obtained by state $X_{i}$ and the FRPDM m using point-to-probability distribution scan matching and feature–point matching. First, the point-to-probability distribution scan matching is performed based on the position and heading predicted by the time update. As a result, we can derive the corrected position and heading from scan matching, which are used as measurement. Another measurement is the range and bearing between the feature points and the position and heading predicted by the time update. $x_{i}$ and $y_{i}$ are mean of the point cloud probability distribution. Using $x_{i}$ and $y_{i}$, we can derive i-th range ($r_{i}$) and bearing ($\beta_{i}$) between probability distribution and predicted vehicle position and heading.
(7)$$H=\frac{\partial h({\hat{X}}_{t}^{-},m)}{\partial {\hat{X}}_{t}^{-}}=\left[\begin{array}{ccc} \begin{array}{c} 1\\ 0\\ 0 \end{array} & \begin{array}{c} 0\\ 1\\ 0 \end{array} & \begin{array}{c} 0\\ 0\\ 1 \end{array}\\ \frac{-(x_{1}-x_{t})}{\sqrt{{\left(x_{1}-x_{t}\right)}^{2}+{\left(y_{1}-y_{t}\right)}^{2}}} & \frac{-(y_{1}-y_{t})}{\sqrt{{\left(x_{1}-x_{t}\right)}^{2}+{\left(y_{1}-y_{t}\right)}^{2}}} & 0\\ \frac{\left(y_{1}-y_{t}\right)}{{\left(x_{1}-x_{t}\right)}^{2}+{\left(y_{1}-y_{t}\right)}^{2}} & \frac{-(x_{1}-x_{t})}{{\left(x_{1}-x_{t}\right)}^{2}+{\left(y_{1}-y_{t}\right)}^{2}} & -1\\ \vdots & \vdots & \vdots \\ \frac{-(x_{n}-x_{t})}{\sqrt{{\left(x_{n}-x_{t}\right)}^{2}+{\left(y_{n}-y_{t}\right)}^{2}}} & \frac{-(y_{n}-y_{t})}{\sqrt{{\left(x_{n}-x_{t}\right)}^{2}+{\left(y_{n}-y_{t}\right)}^{2}}} & 0\\ \frac{\left(y_{n}-y_{t}\right)}{{\left(x_{n}-x_{t}\right)}^{2}+{\left(y_{n}-y_{t}\right)}^{2}} & \frac{-(x_{n}-x_{t})}{{\left(x_{n}-x_{t}\right)}^{2}+{\left(y_{n}-y_{t}\right)}^{2}} & -1 \end{array}\right],$$

In Equation (7), H is the observation matrix for the measurement update, where the observation matrix is derived using the Jacobian calculated by the partial differentiation of all measurements into the state. As described in [8], for the observability of the measurement update, the state estimation can be performed even with only one feature point. In this study, in addition to the feature point, the error correction of the vehicle derived by the point-to-probability distribution scan matching was obtained. Therefore, the vehicle localization can be performed even when there are no feature points.
(8)$$\begin{array}{c} {\tilde{y}}_{t}=z-h({\hat{X}}_{t}^{-},m)\\ K=P_{t}^{-}H^{T}{\left(HP_{t}^{-}H^{T}+R\right)}^{-1}\\ {\hat{X}}_{t}={\hat{X}}_{t}^{-}+K{\tilde{y}}_{t}\\ P_{t}=(I-KH)P_{t}^{-} \end{array}.$$

Equation (8) is the measurement update process, where ${\tilde{y}}_{t}$ is the residual between the measurement and the predicted state. The state and the covariance (P) are updated by the Kalman gain (K). 

### 3.3. Error Correction of Initial Vehicle Position Using the FRPDM

For vehicle localization using a map, an initial position for map matching must be determined. Initial position is the position where the vehicle localization process begins. Generally, the initial position is set based on the position of the GPS/DR. However, in urban areas, map matching may not be performed properly because the position error of the GPS/DR tends to be very large.

Figure 26 shows the error of the GPS/DR in urban areas. The GPS/DR used was a commercial sensor Micro-Infinite CruzCore DS6200. The ground truth is the map trajectory derived from pose graph optimization in Section 2.1. As shown in the Figure 26, in urban areas, big errors tend to occur because of blockage by buildings or multipath. Therefore, it is necessary to correct the vehicle position error for precise map matching. Generally, in the case of vehicle localization using a map, map matching is difficult when the initial position error is large. This is because the chance of proper matching during the data association process is low when the sensor data are far from the map data. In this study, we performed error correction of the initial vehicle position through scan matching using probability distributions of vertical structures. Because the vertical structures tend to be far from one another, it is easy to perform data association. In addition, the outer walls of buildings are usually long. Therefore, the convergence is fast, even if the error is large. Thus, even if the GPS/DR error is large, error correction is possible.

Figure 27 shows an example of the map matching result when the initial position error is large. Specifically, panel (a) shows the result of the error correction with the single-resolution PDM in the 2 m resolution, and panel (b) shows the result of the error correction with the FRPDM. The initial error between the point cloud and the PDM is 10 m for each of the x and y axes. As shown in this figure, because the initial position error is larger than the grid resolution in (a), no matching is made between the point cloud and the probability distribution, so the error cannot be corrected. Therefore, it can be seen that the initial point clouds and the matching results overlap. Conversely, the initial position error is corrected by matching the point clouds and the probability distributions through the data association in panel (b). As described, the initial position error can be corrected through matching with a large size of probability distribution as that of a vertical structure.

Figure 28 shows the flowchart of error correction of the vehicle position. First, the approximate position is estimated by performing scan matching between the point clouds and the probability distributions of the vertical structures. Next, position estimation is performed more precisely through scan matching between the point clouds and the probability distributions of the road markings. However, in most cases, as shown in Figure 26, the initial position error is large, so it is necessary to check whether the position is properly estimated. Therefore, the position estimation result was judged by the ratio between the total point clouds, and the point clouds matched through the data association process in Figure 22.

Figure 29 shows the result of error correction of the initial vehicle position. The blue line represents the position error of the GPS/DR, and the red line represents the initial vehicle position error updated through the process in Figure 28. As shown in this figure, the estimated vehicle position occurred up to about 3 m. Considering that the vehicle position is estimated within 3 m, it can be confirmed that error correction of the initial position can be performed using the FRPDM, despite the large position error.

## 4. Experimental Results

The experiment was performed in the same urban area depicted in Figure 4 and we drove two laps to get the vehicle trajectory. The approximate length of the vehicle trajectory was 5 km. In this paper, localization was performed by post-processing to verify the performance using all LIDAR data (10 Hz). Figure 30 shows the experimental configuration. Where panel (a) is the vehicle platform and panel (b) is the 3D LIDAR specification.

Two cases were compared to confirm the experimental results. One is the case of comparing the vehicle localization results using the single resolution PDM, multi-resolution PDM, and the FRPDM. The other was the case of comparing the vehicle localization using only the point-to-probability distribution scan matching, and the vehicle localization using both the point-to-probability distribution scan matching and the feature–point-based map matching. We analyzed the performance and reliability of vehicle localization using the FRPDM through the two cases of comparisons above.

Figure 31 and Figure 32 show the position and heading errors according to the map types. Only the scan matching measurement was used for vehicle localization. In the case of vehicle localization using the single resolution PDM, a large error occurred within a specific section. The reason was that the error occurred during the time update, and it was not corrected properly. As explained in Section 3.2, the increment of the GPS/DR was used for the time update. However, if the measurement obtained through map matching fails to correct the increment error, the increment error accumulates, and the position error increases.

Figure 33 shows the GPS/DR increment error. As shown in this figure, a large position error of the vehicle localization using the single resolution PDM occurred when a large GPS/DR increment error occurred. Therefore, it is difficult to correct a large error by map matching when using the single resolution PDM, which has small grid resolution. In particular, when the grid resolution is 1 m, it takes a long time to correct the large error because the single resolution PDM cannot match the point and the probability distribution properly if the error is larger than the grid resolution. In contrast, for the FRPDM, even if a large error occurs, the error can be corrected through map matching with a large size of probability distribution.

Table 3 shows the comparison of localization performances among the five types of maps. The grid resolutions of single resolution PDM are 1, 2, and 4 m. Multi-resolution PDM-based vehicle localization is performed for sequential scan matching from the PDM with the largest grid resolution (4 m) to the smallest grid resolution (1 m). As shown in Table 3, in the case of single resolution PDM, the smaller the grid resolution, the higher the RMSE. As shown in Figure 33, if the GPS/DR increment error is large, it is difficult to correct by map matching when grid resolution is small. However, the larger the grid resolution, the worse the matching performance because probability distribution cannot represent the shape of the object properly. In the case of the FRPDM, RMSE and 99% confidence level error of position are smaller than the single resolution PDM. In addition, the position errors of lateral and longitudinal directions must be 0.5 m and 1 m, respectively, at a 95% confidence level for safe autonomous driving. The FRPDM satisfies the condition at 99% confidence level. However, as shown in Table 3, the multi-resolution PDM has the best performance for vehicle localization. The multi-resolution PDM performs scan matching sequentially from the PDM with large grid resolution to small grid resolution. Therefore, it is possible to solve the problem of matching performance according to grid resolution, which is a disadvantage of scan matching using the single resolution PDM. As a result, even if the position error is large, error correction is possible using a PDM with large grid resolution, and precise map matching is possible using a PDM with small grid resolution. However, as shown in Table 4, the multi-resolution PDM has the longest processing time compared to other map types because the multi-resolution PDM is required for scan matching for each grid resolution PDM. In addition, we need to store all of the PDM for each grid resolution. In contrast, the FRPDM performs scan matching only once. In addition, unlike the single resolution PDM, the probability distribution size is variable. Therefore, we can derive fast and precise map matching results using the FRPDM.

Figure 34 and Figure 35 show the positions and the heading errors when only scan matching is used and when scan matching and feature points are used together. As shown in Table 5, using the scan matching and the feature point together demonstrated a better performance to correct the position error in both lateral and longitudinal directions. The heading error was the smallest when using both scan matching and feature points as well. In both cases, the position estimation condition for an autonomous vehicle was satisfied to within a 99% confidence level. However, the reliability of using the feature point was a little higher. When we used the scan matching and the feature-point matching together, errors generated during the scan matching can be corrected by the feature–point measurement. As a result, localization performance is better.

Figure 36 shows the map matching results. Panel (a) shows only the case of scan matching and panel (b) shows the case of scan matching and the feature–point matching together. The blue circle is the ground truth, and the blue star shows the estimated vehicle position. As shown in this figure, the error generated during the scan matching is corrected through the feature–point matching as shown in panel (b). Therefore, using scan matching and the feature–point matching together enables more precise and robust vehicle localization.

Table 6 shows the processing time of vehicle localization using the FRPDM. The algorithm was run based on MATLAB and processed using the CPU only. The CPU was an Intel Core i7-7700 @ 3.60 GHz, and the RAM size was 16 GB. As shown in Table 5, the map matching processing time was 37 ms, indicating that map matching was performed very quickly. The map matching processing time includes the point-to-probability distribution scan matching time, the feature-point matching time, and the EKF execution time. The extraction processing time for the road-marking and vertical structure extraction was 146 ms, so the total processing time was 183 ms. Given that the scan cycle of the 3D LIDAR is 10 Hz, all data cannot be processed in real time on the current algorithm. However, because it was confirmed that map matching can be processed very quickly, accurate and robust vehicle localization is possible in real time, if the point cloud extraction can be processed quickly.

## 5. Conclusions

In this study, we propose FRPDM generation and the FRPDM-based precise vehicle localization method using 3D LIDAR. The FRPDM was generated to be a very small size (61 KB/km). Because it includes road-marking and vertical structure information, it is easily available in urban areas. In addition, because it is a free-resolution map, with no limitation in resolution, small and large probability distributions are included in the same map. Therefore, the convergence is faster than the general point-to-probability distribution scan matching, and the method can perform accurate position estimation. In addition, even if the initial position error is large, map matching is possible through data association using the large size of probability distributions. As a result, error correction of the initial vehicle position is possible. As a result of the FRPDM-based vehicle localization, the RMS position error for the lateral and longitudinal directions are 0.057 m and 0.178 m, respectively. In addition, the RMS heading error is 0.281°. These results satisfy the localization accuracy requirements for autonomous driving. In addition, it is confirmed that the average of the map matching time is 37 ms. As a result, we can estimate the precise and reliable vehicle localization results very quickly using the FRPDM. 

As proposed in this study, we generated a new type of map called the FRPDM and confirmed that robust and precise vehicle localization is possible using the FRPDM. In the future, research on map generation and localization in more areas is required.

## Figures and Tables

**Figure 1 sensors-20-01220-f001:**
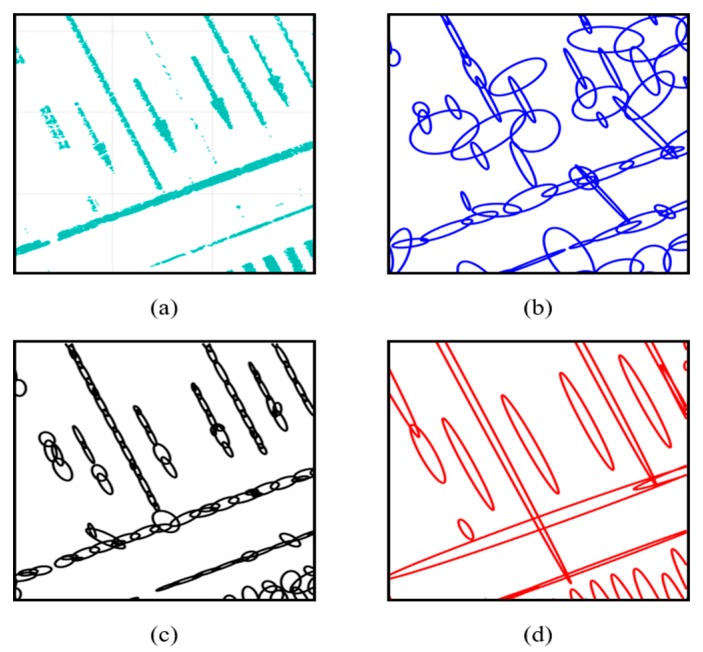
Comparison of probability distribution map (PDM) generation result: (**a**) Point cloud, (**b**) single resolution PDM (2 m), (**c**) single resolution PDM (1 m), and (**d**) free-resolution probability distributions map (FRPDM).

**Figure 2 sensors-20-01220-f002:**
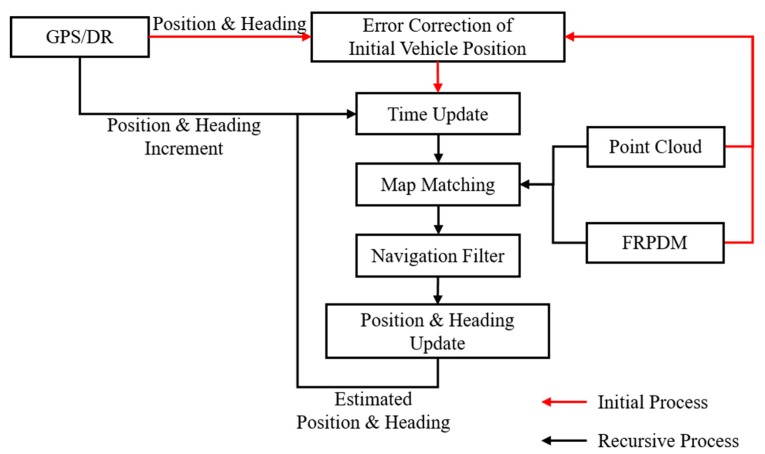
Process of Vehicle Localization.

**Figure 3 sensors-20-01220-f003:**
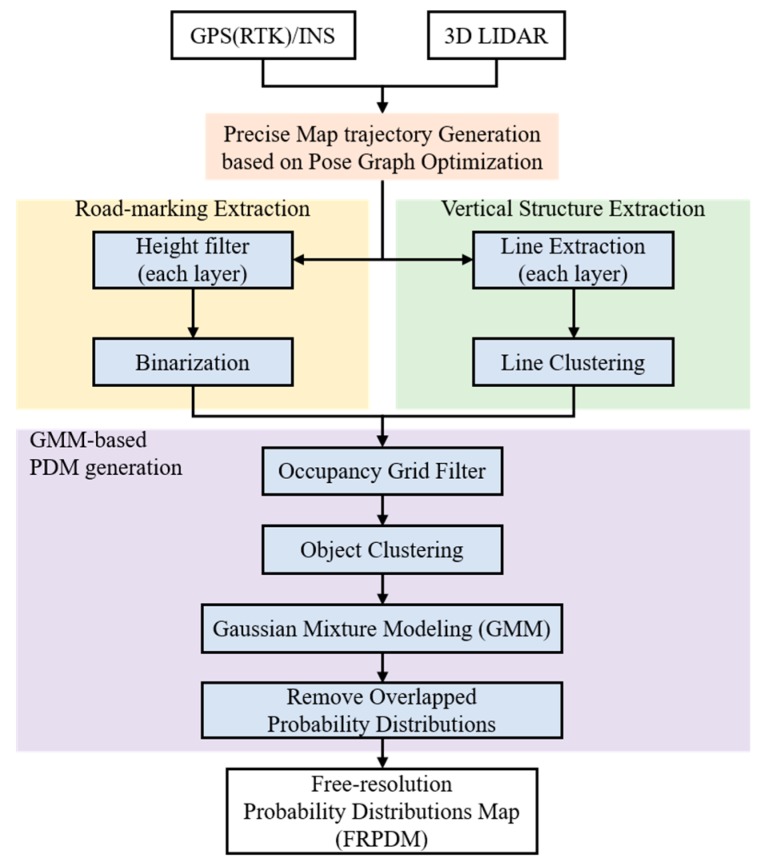
Flowchart of the FRPDM generation process.

**Figure 4 sensors-20-01220-f004:**
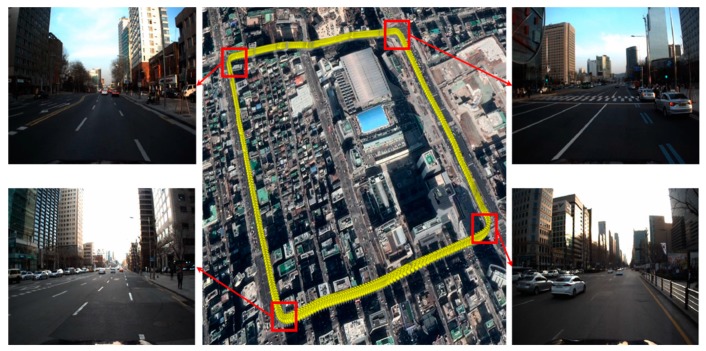
Map trajectory of an urban area.

**Figure 5 sensors-20-01220-f005:**
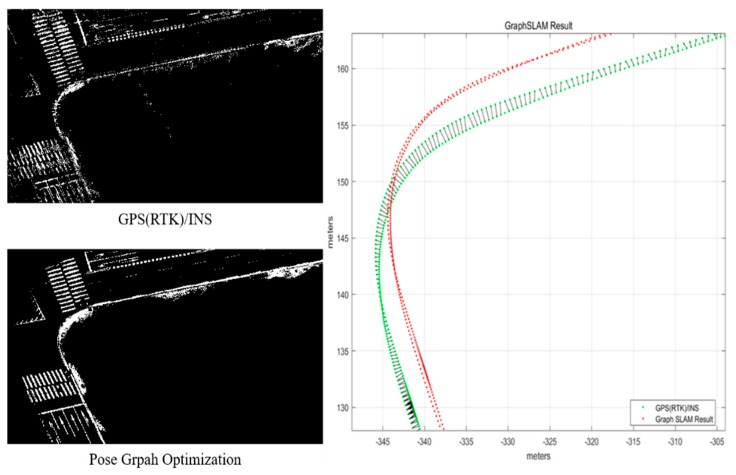
Result of pose graph optimization.

**Figure 6 sensors-20-01220-f006:**
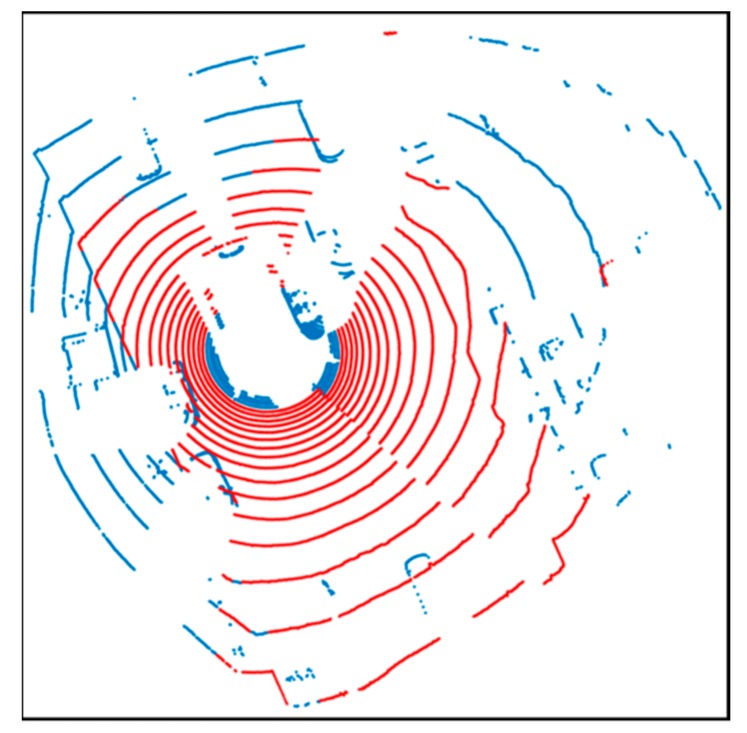
Road surface extraction using height filter.

**Figure 7 sensors-20-01220-f007:**
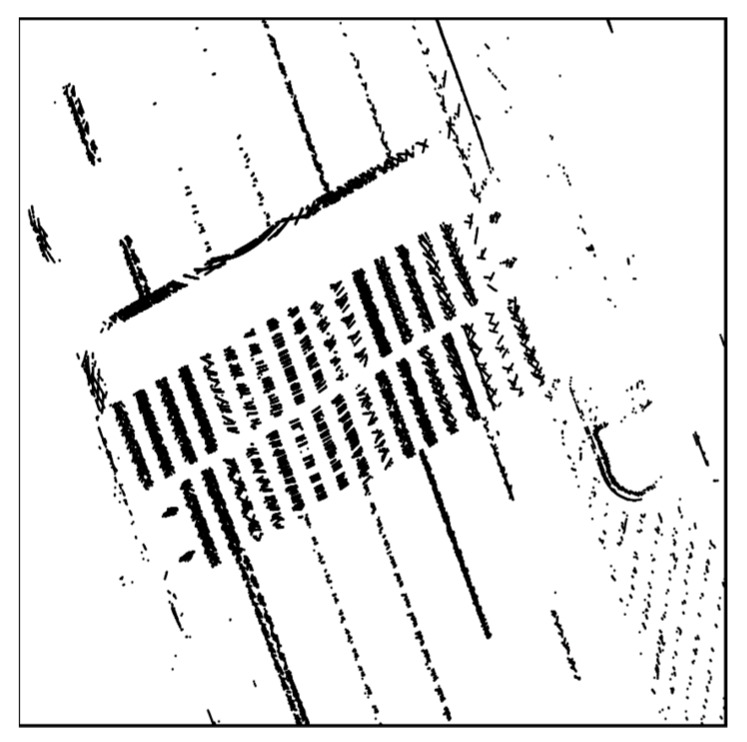
Result of road-marking extraction.

**Figure 8 sensors-20-01220-f008:**
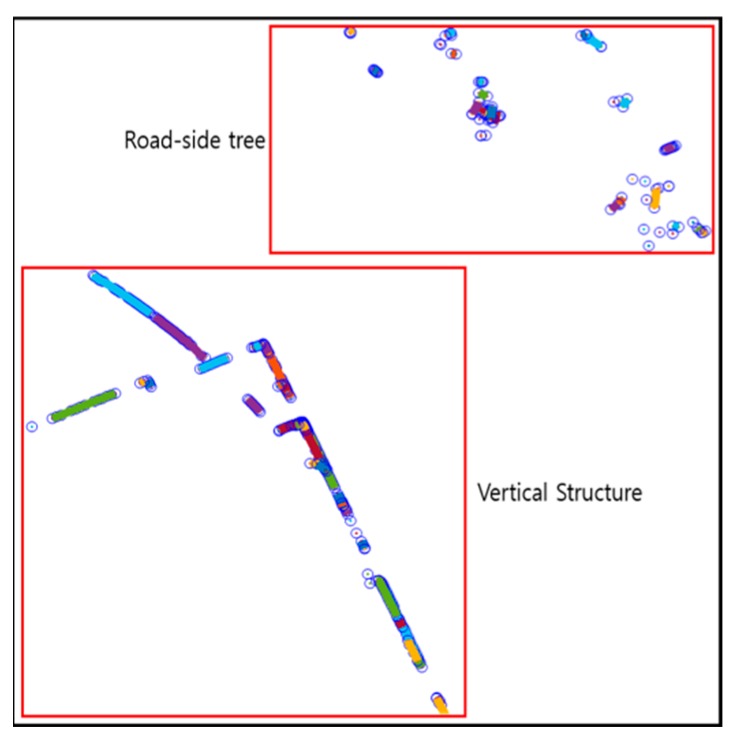
Result of line fitting using iterative-end-point-fit (IEPF) algorithm.

**Figure 9 sensors-20-01220-f009:**
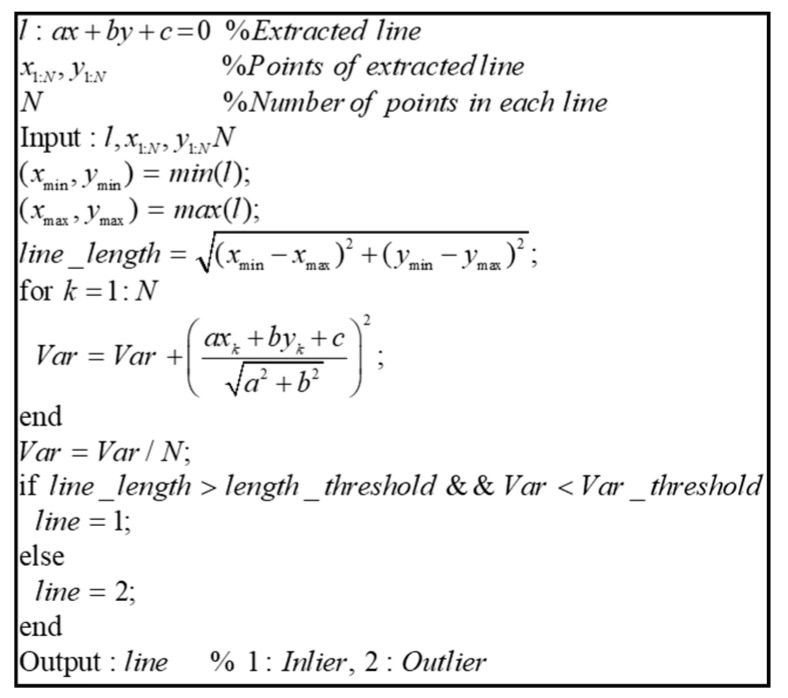
Pseudocode of outlier removal algorithm.

**Figure 10 sensors-20-01220-f010:**
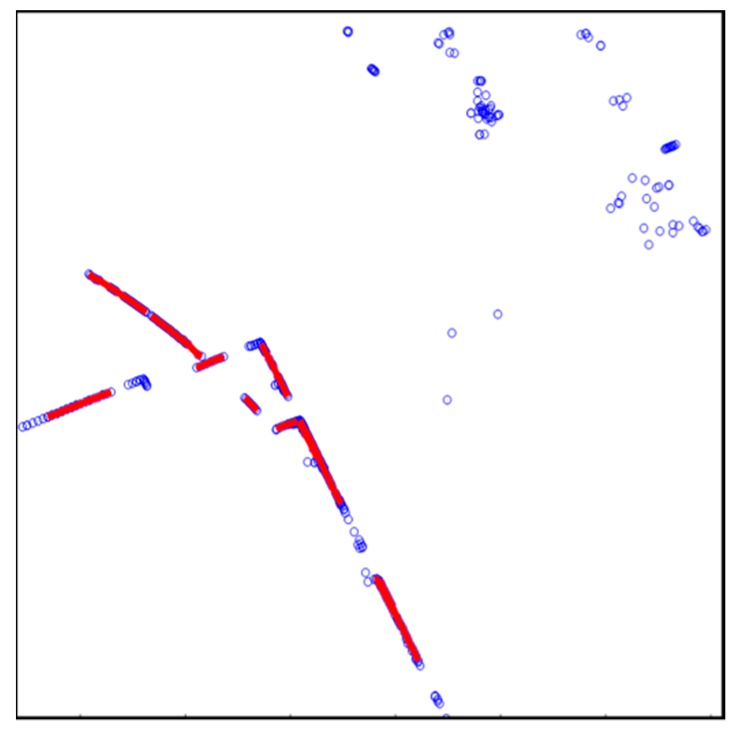
Result of vertical structure line extraction.

**Figure 11 sensors-20-01220-f011:**
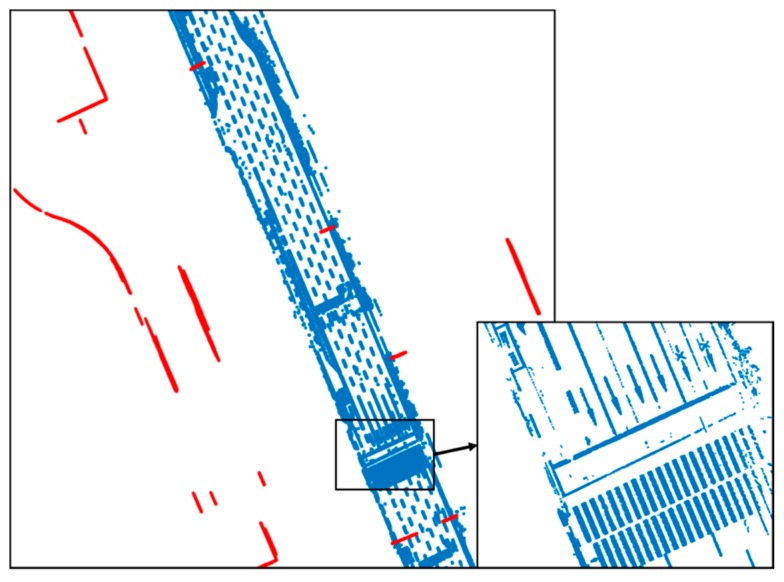
Road-marking and vertical structure extraction results.

**Figure 12 sensors-20-01220-f012:**
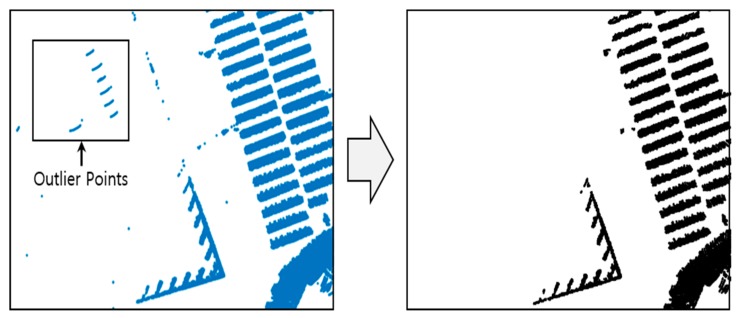
Result of outlier removal using the occupancy grid filter.

**Figure 13 sensors-20-01220-f013:**
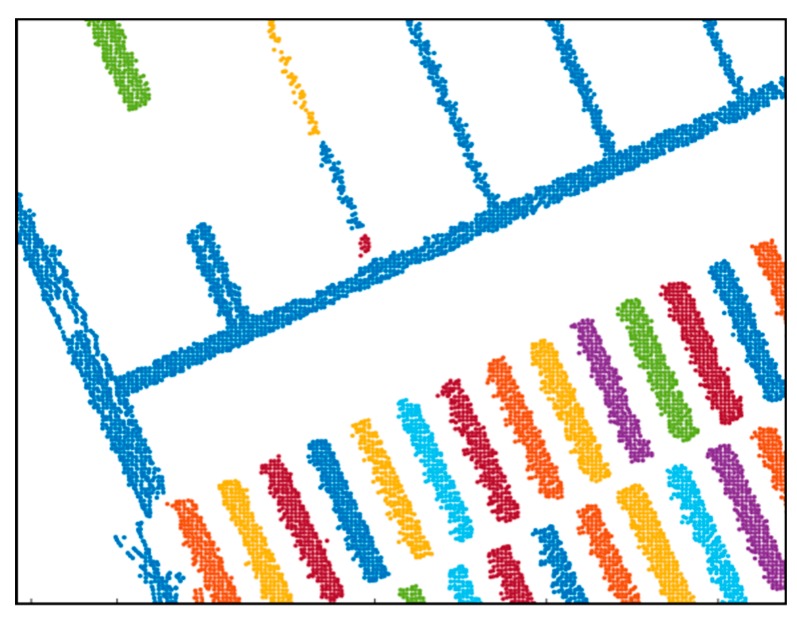
Result of object clustering.

**Figure 14 sensors-20-01220-f014:**
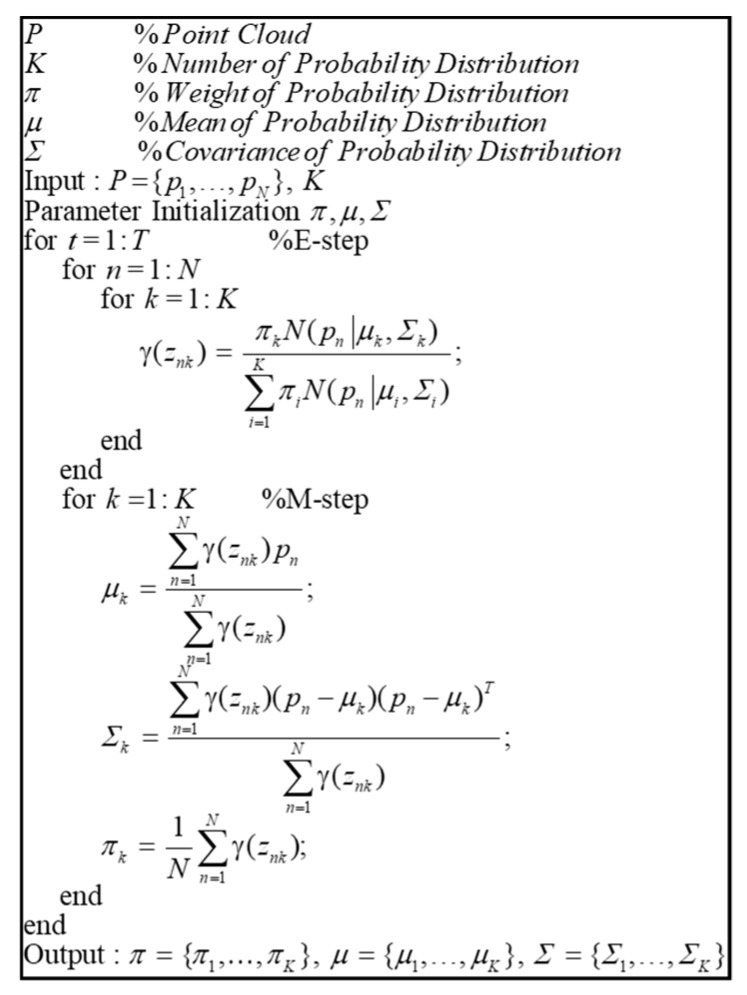
Pseudocode of the expectation–maximization (EM) algorithm for Gaussian mixture modeling (GMM).

**Figure 15 sensors-20-01220-f015:**
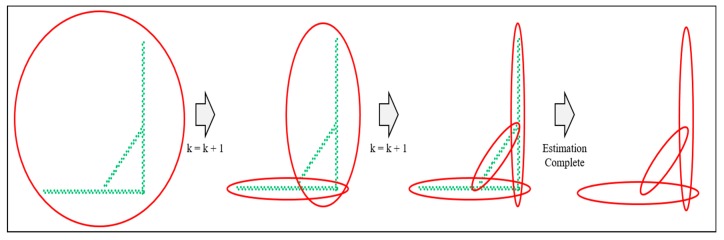
Example of increasing the number of probability distributions.

**Figure 16 sensors-20-01220-f016:**
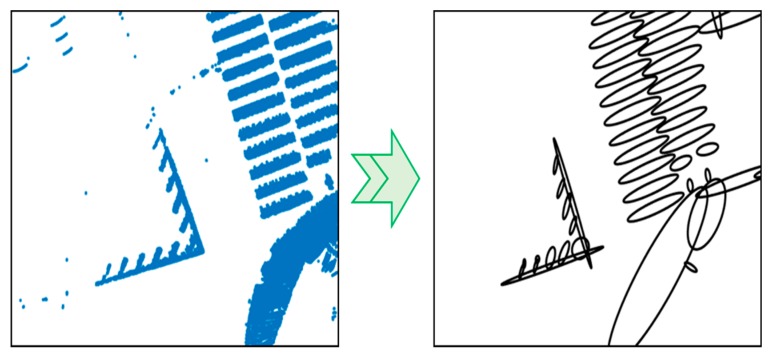
Result of probability distributions transform.

**Figure 17 sensors-20-01220-f017:**
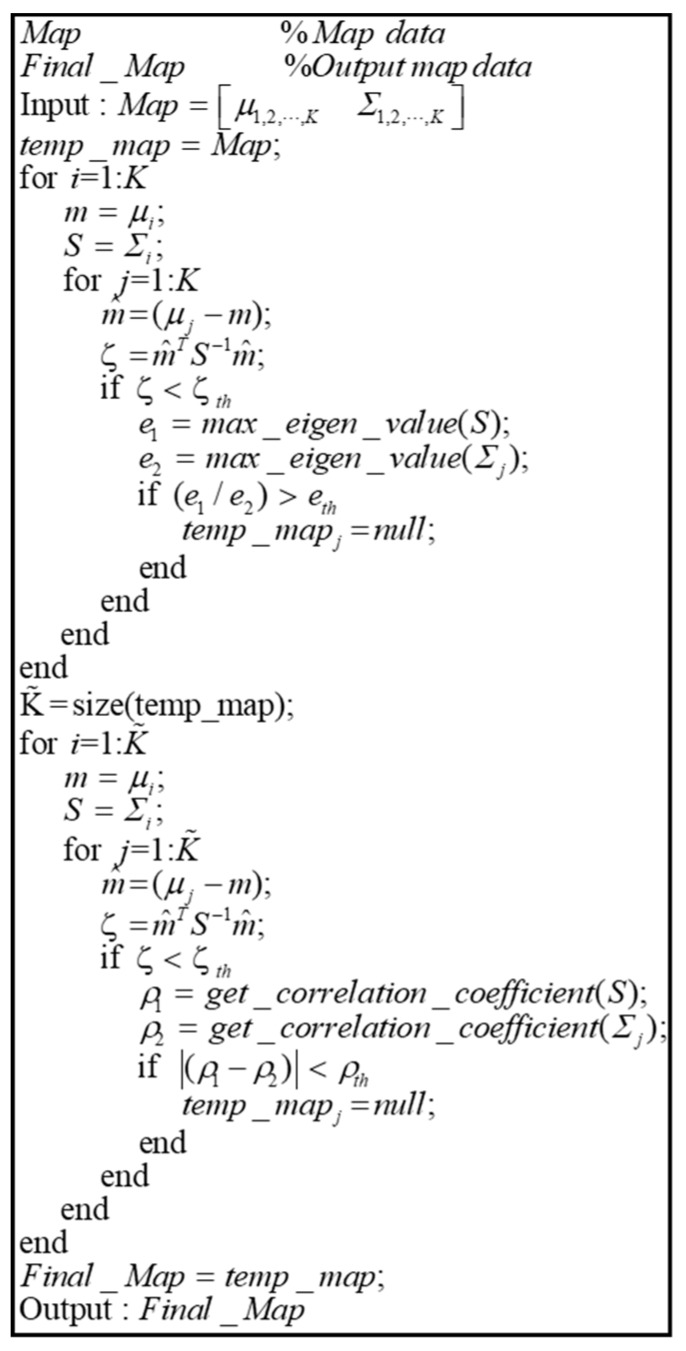
Pseudocode of algorithm for removal of overlapped probability distributions.

**Figure 18 sensors-20-01220-f018:**
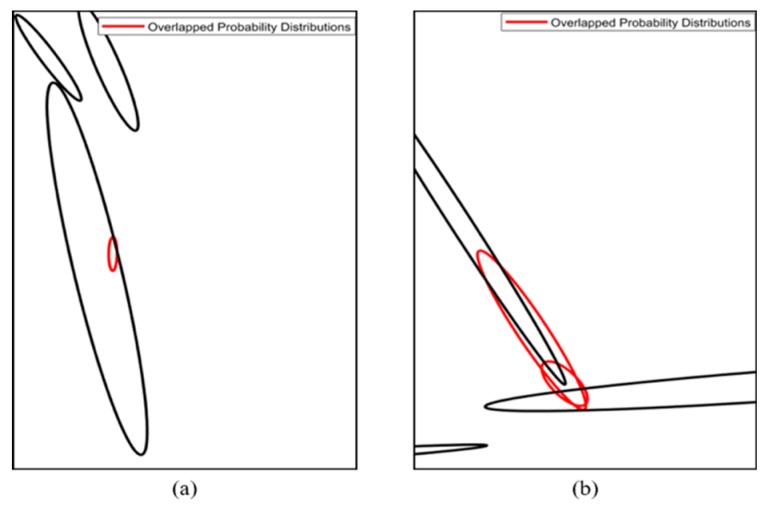
Result of overlapped probability distribution removal method. (**a**) Case of the smaller probability distribution inside a probability distribution, (**b**) Case of the probability distributions overlap with each other.

**Figure 19 sensors-20-01220-f019:**
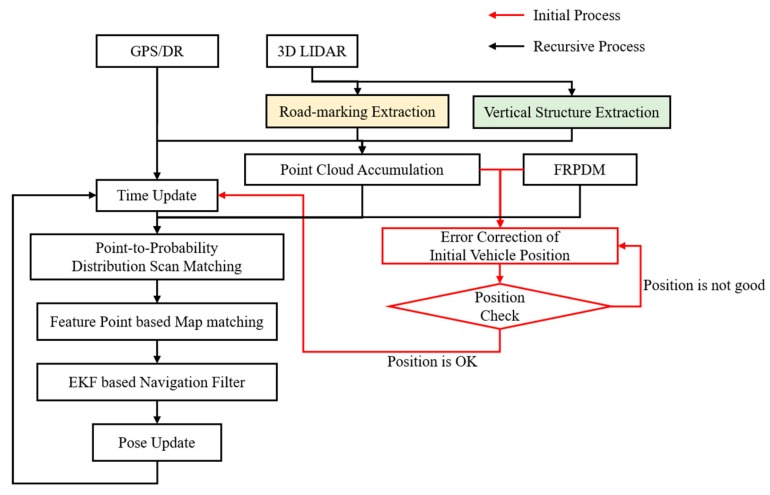
Flowchart of the precise vehicle localization process based on the FRPDM.

**Figure 20 sensors-20-01220-f020:**
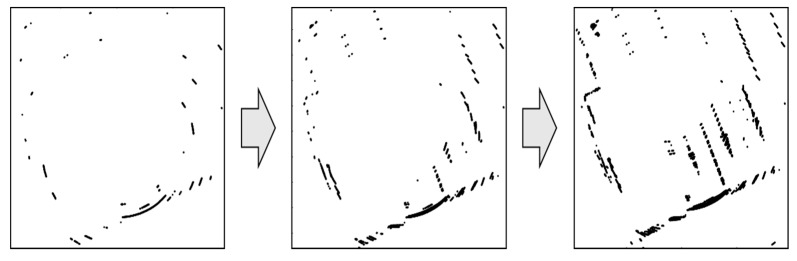
Process of point cloud accumulation.

**Figure 21 sensors-20-01220-f021:**
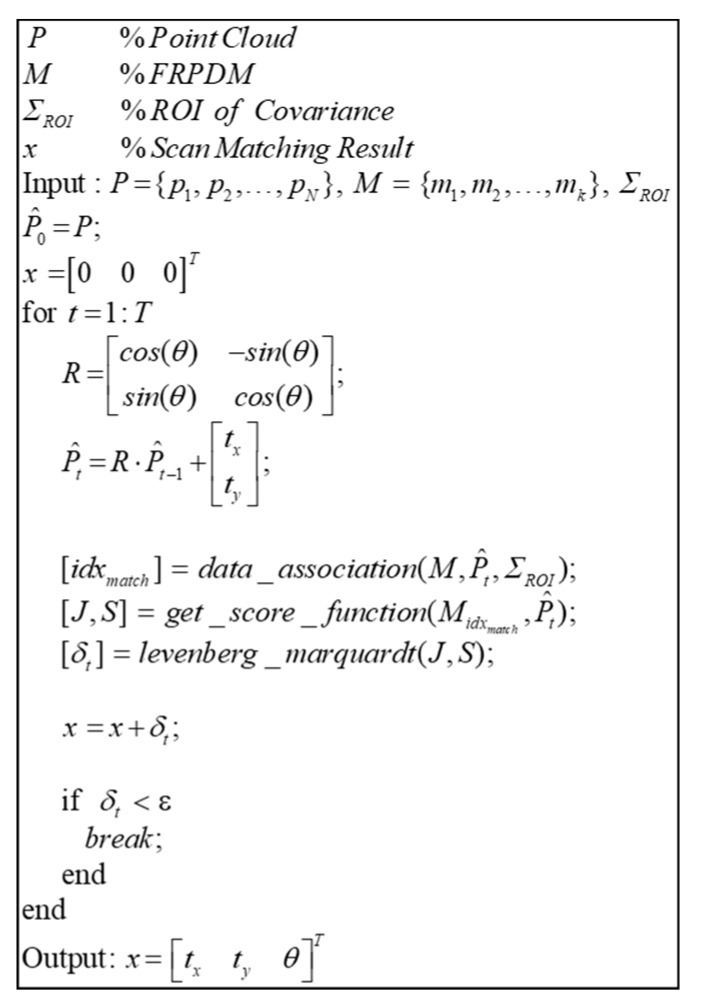
Pseudocode of point-to-probability distribution scan matching.

**Figure 22 sensors-20-01220-f022:**
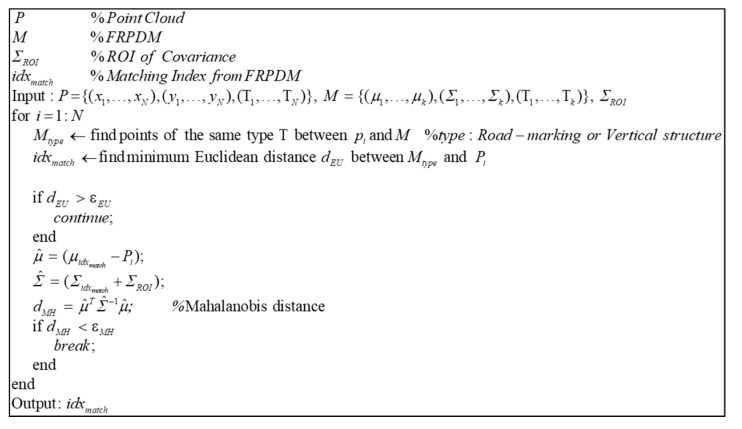
Pseudocode of point-to-probability distribution data association.

**Figure 23 sensors-20-01220-f023:**
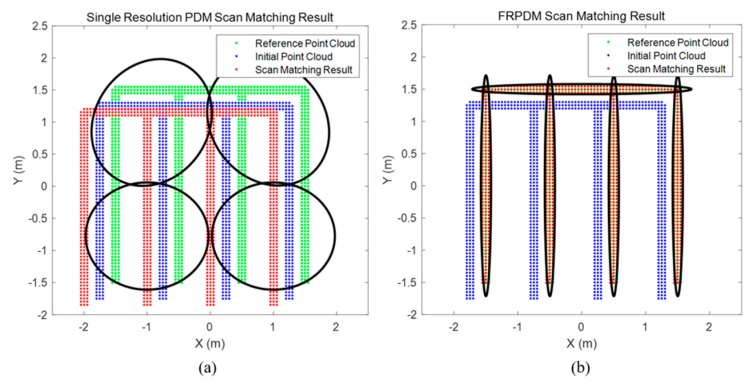
Result of Point-to-Probability Distribution Scan Matching: (**a**) Single resolution PDM (2 m), (**b**) FRPDM.

**Figure 24 sensors-20-01220-f024:**
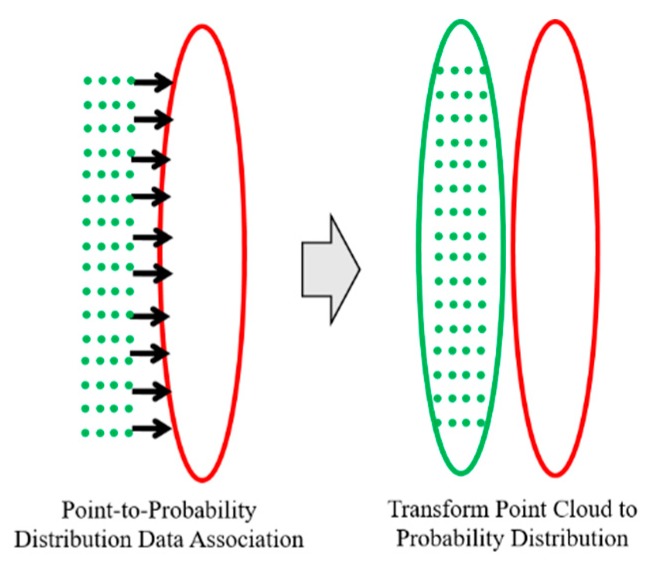
Example of point cloud to probability distribution conversion.

**Figure 25 sensors-20-01220-f025:**
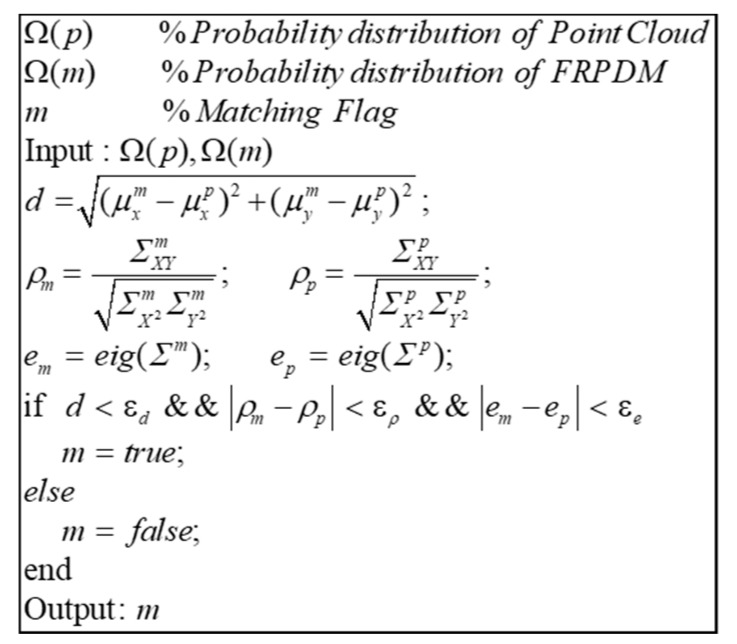
Pseudocode of probability distribution data association.

**Figure 26 sensors-20-01220-f026:**
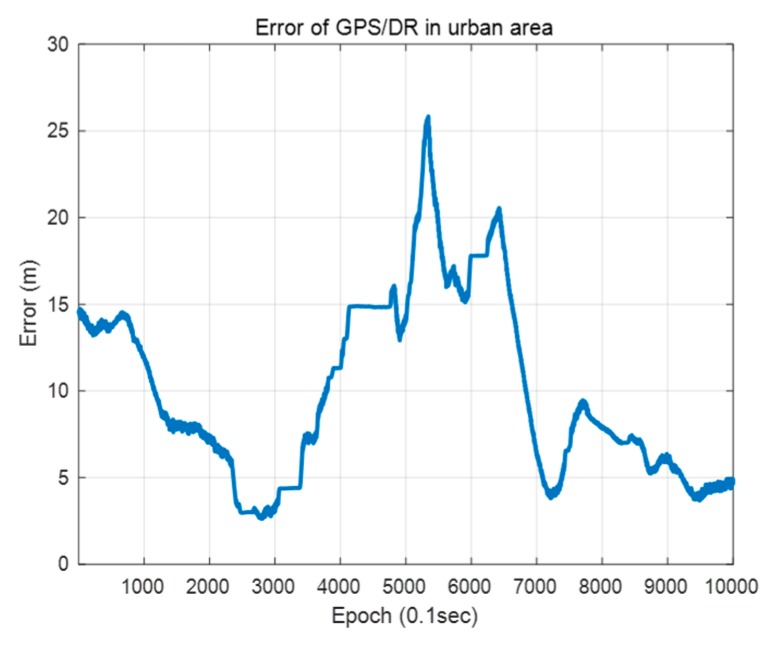
Error of the GPS/DR in urban area.

**Figure 27 sensors-20-01220-f027:**
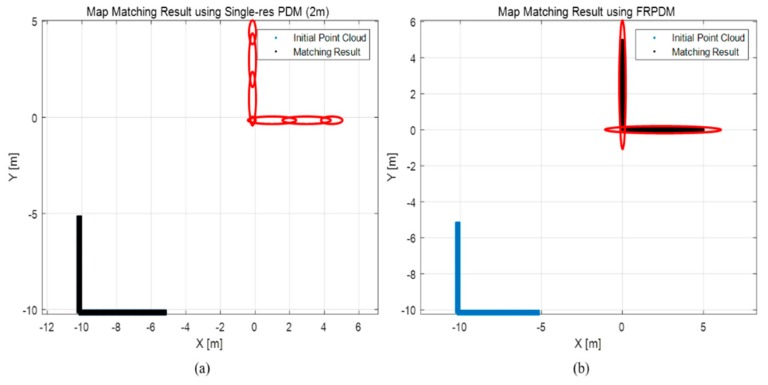
Example of map matching result when initial position error is large. (**a**) Single resolution PDM (2 m), (**b**) FRPDM.

**Figure 28 sensors-20-01220-f028:**
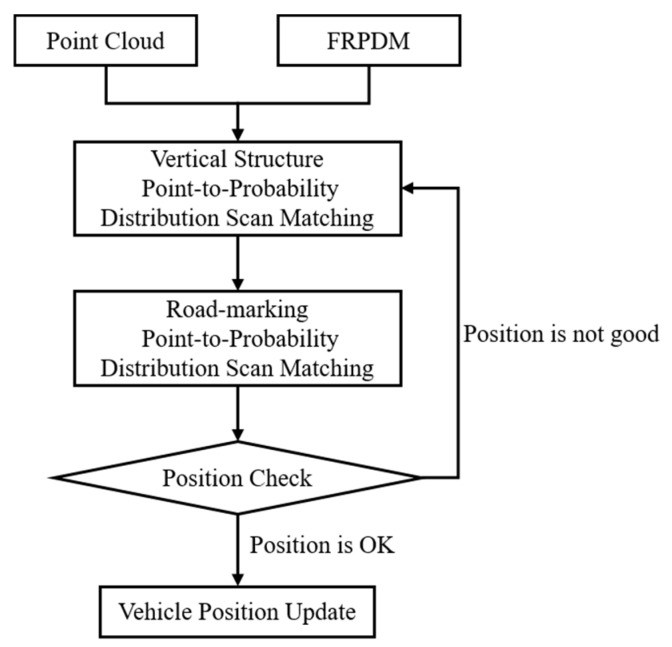
Flowchart of error correction of initial vehicle position.

**Figure 29 sensors-20-01220-f029:**
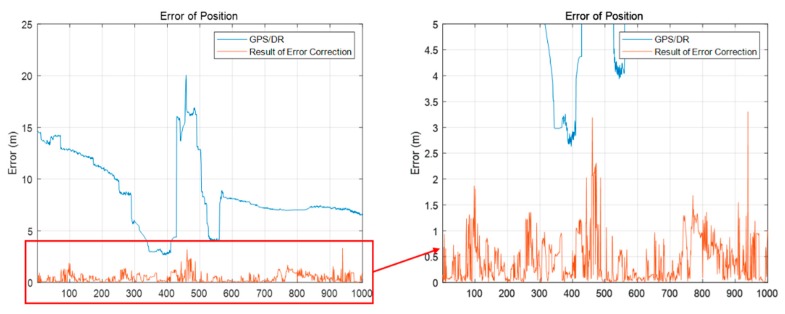
Result of error correction of initial vehicle position.

**Figure 30 sensors-20-01220-f030:**
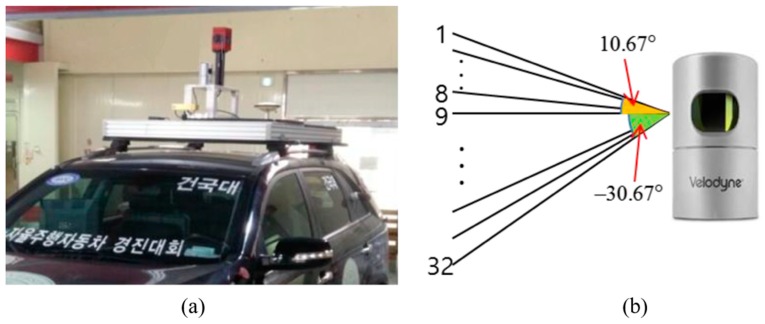
Experimental configuration: (**a**) Vehicle platform, (**b**) 3D light detection and ranging (3D LIDAR) (Velodyne HDL-32E) specification.

**Figure 31 sensors-20-01220-f031:**
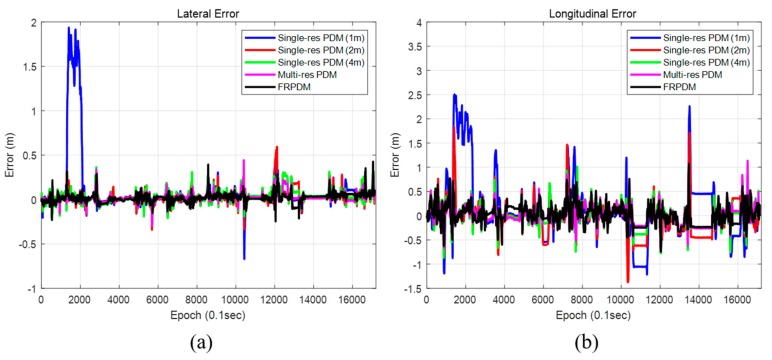
Position error (map type). (**a**) Lateral position error, (**b**) Longitudinal position error.

**Figure 32 sensors-20-01220-f032:**
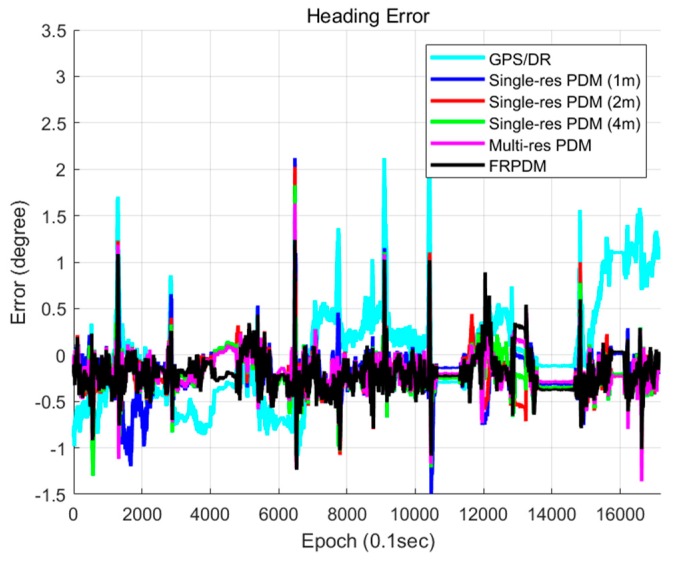
Heading error (map type).

**Figure 33 sensors-20-01220-f033:**
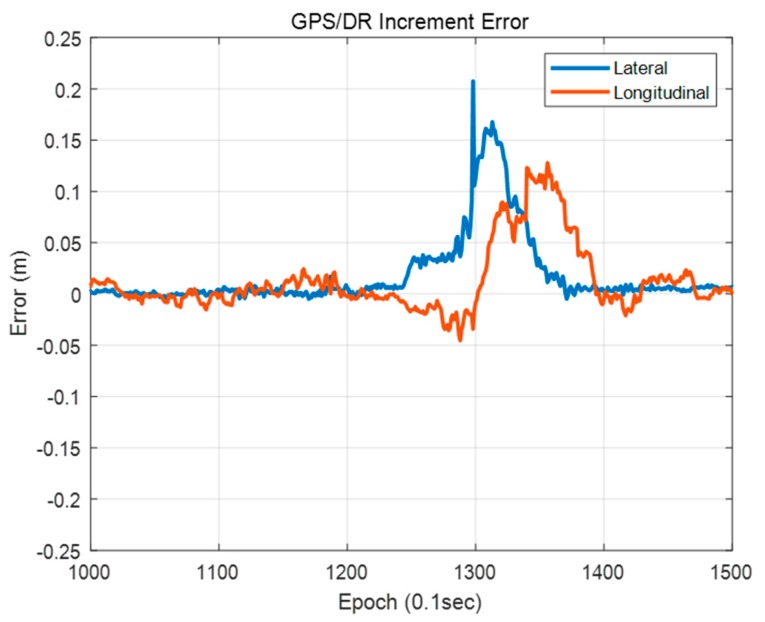
GPS/DR increment error.

**Figure 34 sensors-20-01220-f034:**
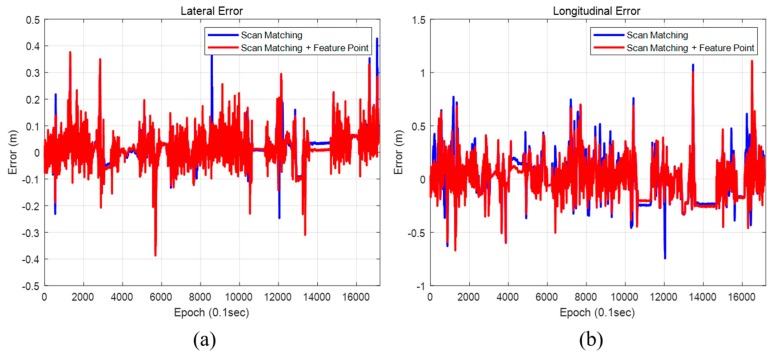
Position error (map matching method). (**a**) Lateral position error, (**b**) Longitudinal position error.

**Figure 35 sensors-20-01220-f035:**
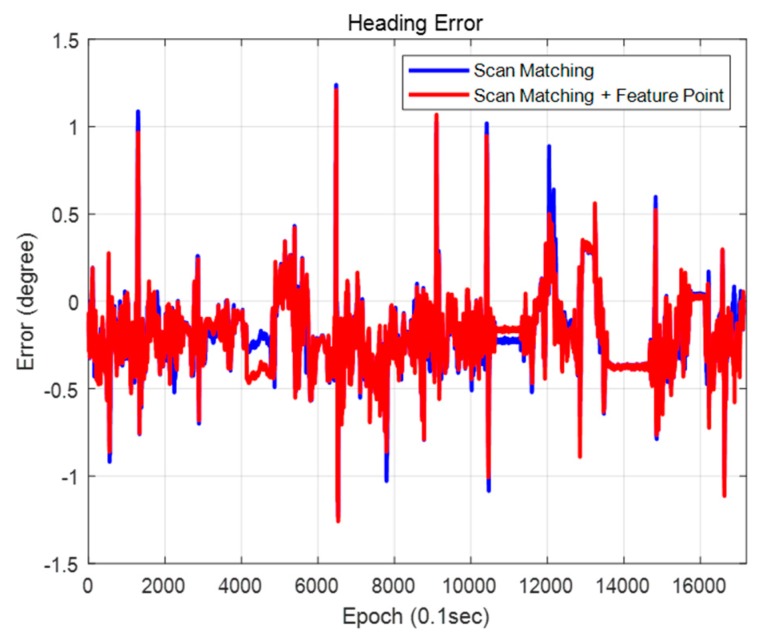
Heading error (map matching method).

**Figure 36 sensors-20-01220-f036:**
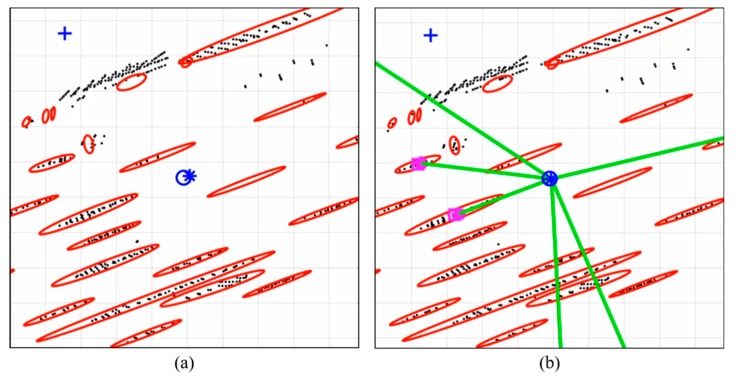
Comparison of Map Matching Result: (**a**) Scan Matching; (**b**) Scan Matching + Feature–Point Matching.

**Table 1 sensors-20-01220-t001:** Example of the FRPDM.

Index	Type	$\mu_{x}$ (°)	$\mu_{y}$ (°)	$\sigma_{x^{2}}$ (m^2^)	$\sigma_{xy}$ (m^2^)	$\sigma_{y^{2}}$ (m^2^)
1	Road marking	37.123521433	127.053851023	1.22	0.43	0.12
N	Vertical Structure	37.143205021	127.050295821	5.78	0.84	0.33

**Table 2 sensors-20-01220-t002:** Comparison of map sizes.

Map Size	Multi-res Gaussian Mixture Map [18]	Binary Grid Map (10 cm) [11]	Extended Line Map [6]	Single-Res PDM (4 m)	FRPDM
**MB/km**	44.3	0.901	0.134	0.124	0.061

**Table 3 sensors-20-01220-t003:** Comparison of localization performances (map type).

Map Type	Position Error (m)				Heading Error (°)
	RMS		99% Confidence Level		RMS
	Lateral	Longitudinal	Lateral	Longitudinal	
Single-res PDM (1 m)	0.336	0.629	1.702	2.182	0.307
Single-res PDM (2 m)	0.079	0.338	0.307	1.326	0.285
Single-res PDM (4 m)	0.074	0.233	0.276	0.736	0.284
Multi-res PDM	0.056	0.189	0.209	0.610	0.257
FRPDM	0.059	0.202	0.234	0.655	0.289

**Table 4 sensors-20-01220-t004:** Comparison of Map Matching Processing Times.

Map Type	Processing Time (s)
Single-res PDM (1 m)	0.039
Single-res PDM (2 m)	0.035
Single-res PDM (4 m)	0.022
Multi-res PDM	0.138
FRPDM	0.037

**Table 5 sensors-20-01220-t005:** Comparison of localization performance (map matching method).

Map Matching Method	Position Error (m)				Heading Error (°)
	RMS		99% Confidence Level		RMS
	Lateral	Longitudinal	Lateral	Longitudinal	
Scan Matching	0.059	0.202	0.234	0.655	0.289
Scan Matching + Feature Point	0.057	0.178	0.219	0.570	0.281

**Table 6 sensors-20-01220-t006:** Processing time of vehicle localization using the FRPDM.

Processing Time (s)		
Extraction	Map Matching	Total Processing Time
0.146	0.037	0.183

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
