# Peer review of "Free-Resolution Probability Distributions Map-Based Precise Vehicle Localization in Urban Areas"

_sensors, 2020, doi:10.3390/s20041220_

Round 1

Reviewer 1 Report

A few grammatical errors i. e. "point matching were performed for obtain the measurements", "We perform map matching", "Table 3 is comparison of".

Author Response

I revised all grammatical errors mentioned by reviewer. Also, awkward grammar and expressions were corrected.

Reviewer 2 Report

This paper proposed a free-resolution probability distribution map and FRPDM based precise vehicle localization method using 3D LIDAR. The results denoted that the root mean square position error and the RMS heading error will reduce. However, the innovation of proposed algorithms including EKF filter is questionable, and there is no comparable results to verify performance advantage.

Reviewer 3 Report

Introduction

Page 2. Lines 44-47. I partially disagree with this statement. Although, line-based maps may be relatively coarse, line information can be used to correct the vehicle localization solution, thus improving its accuracy level. See for instance the reference below

Opromolla, R., Fasano, G., Grassi, M., Savvaris, A., & Moccia, A. (2017). PCA-based line detection from range data for mapping and localization-aiding of UAVs. International Journal of Aerospace Engineering2017.

Page 2. The overview of related work, can be further improved by adding more recent works on this topic. See the examples below

Chen, L. H., & Peng, C. C. (2019). A Robust 2D-SLAM Technology With Environmental Variation Adaptability. IEEE Sensors Journal19(23), 11475-11491.

Li, X., Du, S., Li, G., & Li, H. (2020). Integrate Point-Cloud Segmentation with 3D LiDAR Scan-Matching for Mobile Robot Localization and Mapping. Sensors20(1), 237.

Page 3. Figure 1. Though this figure provides only a qualitative comparison, I would like to suggest the authors to add more information. In this respect, I have the following remarks

Add the multi-resolution grid case. Add information about typology of environmental features identified by the Probability Distributions.

Method of FRDPM generation

Page 10. Could you clarify how the parameters associated to the PD (i.e., weight, mean and covariance) are initialized to apply the pseudocode depicted by Figure 14?

Page 11. Equation 1. A definition of e1 and e2 is missing

Page 13. Lines 347-349. How is the mentioned threshold assigned?

Page 13. Table 1. Units of measurements are missing

FRPDM based precise vehicle localization method

Page 21. Figure 26. Please clarify which is the reference position solution adopted to estimate the GPS positioning error.

Experimental results

Page 23. A detailed description of the experimental setup including all the adopted instruments and the vehicle on which they are installed for data collection is missing. Also, the authors should specify at the beginning of the section that the data are processed off-line as this aspect is not clear till the end of the manuscript.

Conclusions

Pages 27-28. To be fair, the authors could mention that, although the proposed method demonstrated to be the fastest and most memory efficient, the multi-resolution method provided slightly better localization accuracy. Also, a possible future research effort could be the implementation of the method with a different computing language (e.g., C or C++) to verify the possibility of real time implementation.
